# Assessment of the Effect of Secondary Fixation on the Structure of Meat Products Prepared for Scanning Electron Microscopy

**DOI:** 10.3390/foods9040487

**Published:** 2020-04-13

**Authors:** Hana Běhalová, Bohuslava Tremlová, Ludmila Kalčáková, Matej Pospiech, Dani Dordevic

**Affiliations:** 1Department of Plant Origin Foodstuffs Hygiene and Technology, Faculty of Veterinary Hygiene and Ecology, University of Veterinary and Pharmaceutical Sciences Brno, 612 42 Brno, Czech Republic; tremlovab@vfu.cz (B.T.); lida.anezka.l@gmail.com (L.K.); pospiechm@vfu.cz (M.P.); dordevicd@vfu.cz (D.D.); 2Department of Technology and Organization of Public Catering, South Ural State University, 454080 Chelyabinsk, Russia

**Keywords:** osmium tetroxide, scanning electron microscopy, lipid droplets, artifacts, meat

## Abstract

The aim of the research was to verify the necessity of secondary fixation with osmium tetroxide in various types of meat products and evaluation of structural changes of products using different fixation procedures. The material for the study consisted of 11 types of meat products that were analyzed using a scanning electron microscope (SEM) with two different methods of chemical fixation. The first method included the usual processing of biological samples: glutaraldehyde primary fixation, the use of a buffer, secondary fixation by osmium tetroxide (OsO_4_), buffer, and dehydration using ethanol of increasing concentrations. The second method comprised the glutaraldehyde primary fixation and dehydration using the ethanol of increasing concentrations only. The results unambiguously suggest that the main difference between these methods is in fixation and visibility of fat. Our analysis principally suggests that fixation of the product with OsO_4_ allows the tracking of all components (fat droplets, muscle fibers, connective tissue) in meat products. At the same time, our results also support the possibility that the secondary fixation can be skipped during the analysis, where the main objection is an observation of lipid-free structures of the meat products (e.g., connection between muscle and starches or spices) or meat products with an insignificant amount of fat.

## 1. Introduction

Meat and meat products are broadly consumed in the Czech Republic. In 2017, the consumption of meat and meat products was 80.3 kg per person, and in the year 2018, this number slightly increased [1] which is well above the EU average consumption—64.8 kg per person [2] and more than double the consumption compared to the world—30.6 kg [3].

The Czech legislation comprises Decree No. 69/2016 Coll. on requirements for meat, meat products, fishery and aquaculture products, and products thereof, eggs and products thereof, which apart from the basic definitions, composition and sensory requirements, divides meat products into two main groups—meat products and semi-finished meat products. The meat products are further divided in seven subgroups: heat-treated, non-heat treated, non-heat treated for heat processing, long-life heat-treated, long-life fermented, cans, and semi-preserves [4].

The fact that meat products are widely popular in the Czech Republic plays into the hands of some of the producers, who make efforts to exchange more expensive components for cheaper substitutes. This kind of tampering can be fought by knowing the structures of the individual components, the way they are connected to each other in a product or by knowing the structure of the whole product. The structure of a product can be studied using both the light and electron microscopy. The indisputable advantage of the scanning electron microscopy (SEM) is its ability to show the surface of a product [5] and its microstructure in a 3D imaging [6]. Not only does it give us the idea about individual components, but also the connection between them. Such facts may help us when proving product tampering when studying the behavior of the individual components of a product during technological processing or with the development of new additives. 

To be able to study the structure of meat products using SEM, they must be prepared as every other biological sample—by fixation [7]. Fixation aims to maintain the sample as close to the native state as possible [8]. On the other hand, every step may lead to damage to the final sample, for example, by changing the volume of the sample [9,10] or by the osmotic pressure effect on the cells [11]. That is why the goal is to adjust fixation to the given sample so that it includes the least steps while providing a sufficient fixation [12]. The common procedure of fixation of biological samples includes primary and secondary fixation, dehydration in ethanol of increasing concentrations, and drying [13]. The primary fixation is carried out using glutaraldehyde or formaldehyde. These aldehydes stabilize all components of cell membranes, especially proteins [14]. However, glutaraldehyde reacts poorly with lipids. Therefore, it is good to add a secondary fixation using osmium tetroxide, which stabilizes especially lipids. However, during the long-term exposition, the osmium tetroxide may damage the proteins that are already fixated and other components of the membranes. Therefore, fixation with OsO_4_ is recommended only when necessary [15]. 

Our study aimed to verify the necessity of secondary fixation with osmium tetroxide in various types of meat products and to assess structural changes in the products when using different methods of fixation. 

## 2. Material and Method

The selection of samples obeyed Decree No. 69/2016 Coll. [4], on requirements for meat, meat products, fishery and aquaculture products, and products thereof, eggs, and products thereof. The decree categorizes meat products into 7 groups. The semi-finished processed meats are a standalone group. Two products were selected from each group of meat products, except semi-preserves for which we only found one product. The following products were analyzed: heat-treated—English bacon, Orava bacon; non-heat treated—“Čajovka”, “Métský” salami; long-life heat-treated—“Vysočina”, “Tourist” salami; long-life fermented—“Dunajská” sausage, “Poličan”; cans—Játrovka”, “Matěj”; semi-preserves—Moravian sausage mix. 

Samples of the selected meat products were obtained from a chain of markets. The bought products were adjusted to the appropriate size (1 cm × 1 cm × 0.5 cm). Four samples were randomly selected from each product for each procedure, following the regulations for sample harvesting according to Decree No. 231/2106 Coll. [16]. The samples were subsequently processed using various methods, see Table 1. Further processing was the same for all samples regardless of the different types of chemical fixation. The samples were dried using the CPD method (critical point drying) using the Emitech K850 device (Quorum Technologies, Laughton, United Kingdom). The next step consisted of sticking the samples to metal plates using double-sided adhesive tape and followed by coating with a 10 nm layer of gold using the sputter coater device Q150R ES (Quorum Technologies, Laughton, United Kingdom). The resulting pictures were taken by the scanning electron microscope MIRA3 (Tescan, as., Brno, Czech Republic), under adequate conditions (Figure 1).

## 3. Results and Discussion

The results are summarized in Table 2. At first glance, the table shows that not all the meat products were suitable for processing using the SEM method. Both methods of processing seemed to be unsuitable for the non-heat treated meat products (“čajovka” and “métský” salami). All findings from Table 2 are explained additionally in Figure 2, Figure 3, Figure 4, Figure 5, Figure 6, Figure 7, Figure 8 and Figure 9. 

The captured structure can be seen in Figure 2. The final structure of the product was the same for both procedures and both products. It is apparent, that the individual components of the product were not recognizable in the sample, even though the manufacturer mentions on the label that “čajovka” contains 70% and the “métský” salami 31% of meat. The possible explanation could be the technological processing. The strong and thorough grinding of the components and subsequent whisking with air can lead to creating emulsion or foam [17]. It was difficult to work with the samples—they could not be sliced, they were soft to touch, whisked, rather than mashed. The primary and secondary fixation did not affect them, the samples remained soft, even after the fixation and subsequent processing, which is contradictory to Kiernan [18] or Nebesářova [19], who suggested that the use of glutaraldehyde and OsO_4_ mechanically solidifies surface structures. Because the samples were not solidified, it led to insufficient adhesion on the adhesive tape and physical shifting during the microscopy [20]. The cryo-SEM method seemed to be better for samples such as these, as it shows fine structures, such as emulsions or foams better [21].

The other samples became harder after the primary fixation, and they were easier to manipulate. Samples representing the cans group were similar in consistency (before starting the fixation) (Figure 3), but without the issues that the non-heat treated products had. In both samples from the can group, certain structures were visible. Areas of visible muscle strands could be found (both cans state a certain percentage of muscle in the composition (Figure 3d). “Játrovka” also stated 26% content of liver; however, they were not found when we studied the structure, which can be due to thorough grinding of the ingredient. There was a visible difference between the representatives of cans in the presence of fat droplets, where the second procedure without secondary fixation by OsO_4_ showed only empty spaces that did not contain fat droplets. For the other parameters, the method of processing did not play any role. Other findings that were not impacted by the method of fixation were air gaps, modified starch, or crosslinked protein (Figure 3a), probably released from the muscle during technological processing. 

Two whole-muscle products were chosen as representatives from the group of heat-treated meat products—English bacon and Orava bacon. In both products, there was a visible difference in structure compared to other samples—these products were made from one piece of muscle; therefore, the muscle fibers were clearly visible regardless of the chosen method of processing (Figure 4). In both cases, we could also see connective tissue that surrounds and connects the muscle fibers into larger units—muscle fascicles (Figure 4d,f–h). The main difference, again, was in the presence of fixated lipid droplets in the fatty tissue or connective tissue, which were visible during the work procedure 1 with secondary fixation (Figure 4a–d). Figure 4e,g show empty gaps in the connective tissue, which were demarcating the lipid droplets. The other difference was in the presence of artifacts during procedure 1 (Figure 4a), which occurred during the processing and did not belong to the structure of the product. 

The two most favorite salami in the Czech Republic—the “Vysočina” and “Tourist” salami were chosen as representatives from the group of long-life heat-treated meat products. These products are favored for their taste and shelf-life. The main component in both cases is pork meat, beef, and lard. The muscle was visible in both samples regardless of the method of processing. We were not able to determine the origin of the given muscle; however, that was not the intention of this study. Fat, and hence, the lard, was visible at the first method of processing (Figure 5a–d). The fat at the sample was fixated in the original places, but also, in both samples, you can see the fat that has “spilled” and covers other structures of the sample (Figure 5aD,d), which can be considered for an artifact [22]. In both procedures, air gaps were found (Figure 5g,h) emerging during stirring while being manufactured. Procedure 1 yielded artifacts (Figure 5a,d) that occurred during processing in the lab due to being exposed to chemicals [23]. The occurrence of artifacts has been described by many authors. Hopwood credits glutaraldehyde for their formation [24]. Gil and Weibel credit the combination of glutaraldehyde and osmium tetroxide under the influence of phosphate buffer [25]. These artifacts occurred in our work mostly in samples processed by the first procedure, and so it may be presumed that their formation is probably affected by the cacodylate buffer or osmium tetroxide, which is contradictory to the study of Hopwood, who described the formation of artifacts after the exposition to glutaraldehyde [24]. The authors of this study explain this by the difference in method. In view of the fact that the chemicals can be stored for a certain amount of time, these artifacts might have possibly occurred due to the use of old or poorly stored solutions [19]. Therefore, it is advisable to always prepare fresh fixative solutions [20]. Nebesářová does not recommend the use of a fixative solution of OsO_4_ when it changes color (from light yellow to clear) [19]. The assessment of color by the naked eye is, however, highly subjective [26]. Hence, this condition appears to be unsatisfactory. Figure 5g,h shows Tourist salami processed by procedure 2. The picture clearly shows that the fixation and processing was unsatisfactory; the samples were as if covered by a layer of metal, and no structures were apparent, even when using the same chemicals and same procedure as with the sample of “Vysočina”. This proves that the processing of biological samples (including food) is considerably difficult and very challenging. 

Among the other analyzed products were “Dunajská” sausage and “Poličan”. Both products belong to the group of fermented long-life meat products. An added starter culture was specific for these products, usually lactic acid bacteria (LAB). These are added to the product for their ability to lower the pH of the product and the activity of water (a_w_), which contributes to longer shelf-life even without a heat treatment [27,28]. We managed to capture these bacteria in both of the samples (both the “Dunajská” sausage and “Poličan”) in both methods (Figure 6). Figure 6c shows the arrangement of LAB in the product. We can presume that the method of processing the sample did not affect the cell structures of LAB. According to the morphology of the bacteria, it can be concluded that it was most probably *Lactobacillus plantarum* in the “Dunajská” sausage [29] and bacteria of the *Pediococcus* species in “Poličan” [30]. Both bacteria are commonly used as starter cultures in meat products [31], and they morphologically correspond with our findings. Table 2 shows that the visible part of muscle, (Figure 7c,d,g), which was the main component, could be found in both products and both procedures. Again, the only difference was in the presence if fixated fat, which was present in both samples processed by the first method (Figure 7a,d). The second difference was in the number of artifacts, which were greater in samples processed by method 1 (Figure 7a). In “Dunajská” sausage, we also managed to capture spices, specifically garlic (stated in the composition) and paprika (Figure 9c), which were not stated in the composition, but it may be expected that it was a part of the seasoning mix. 

The group of semi-preserves is the last group of meat products (Figure 8). The only representative that we managed to get in the Czech markets was the Moravian sausage mix. The semi-preserved contained more than 90% of meat; therefore, you could mainly see muscle in the samples (Figure 8c). Another significant finding was spices, namely paprika, caraway, and mustard (Figure 9a,b). Among the procedures, there was a difference in the presence of fat and artefacts (Figure 8a,c). 

The above-mentioned shows, that the most considerable difference between the procedures was in the presence of fat in the form of fixed lipid droplets due to the ability of OsO_4_ to fixate lipids. However, OsO_4_ is used within the electron microscopy for many reasons. Osmium (Os), in its original state, is a conductive metal. This quality is also passed to its compounds. A sample fixed using the OsO_4_ does not have to be metal-coated in certain cases; thus, it is not necessary to apply another layer of metal, which may prevent the covering of the very fine structures on the surface of the sample [32]. In addition, during the typical procedure in which the sample is coated with a layer of metal, the O_s_O_4_ helps to lower the charge of the sample [33,34] (light areas on Figure 2a,e,g). Different kinds of electrons are used during the scanning electron microscopy. The secondary electrons are important for the formation of the image as they have low energy and are slower than the primary electrons. That is why it is important to remove these primary electrons from the surface of the sample as fast as possible using the conductive metals so that they would not spoil the detection of the secondary electrons and, subsequently, the formation of the image [19,35]. We only managed to capture this difference in the group of cans, where neither of the samples processed by method 2 showed apparent structure, and the samples got charged—light areas in Figure 3e–g. We did not manage to confirm this fact in the other samples, as the difference among the procedures did not show, just as the increase in contrast on the final image due to OsO_4_ [20,36,37] was not noticeable.

## 4. Conclusions

In summary, in this paper, we analyzed 11 meat products that were made in the Czech Republic. Our findings principally suggest that heat non treated meat products are not suitable for processing using SEM, presumably due to their paste-like structures. Second, our analysis supports that the main difference between the two fixation procedures was the presence of visible fat, which leads the authors to the conclusion that the secondary fixation using OsO_4_ affects the fixation of lipids in the product. Furthermore, our findings also support that the secondary fixation can be skipped if the objective is to observe lipid-free structures of the meat products (e.g., connection between muscle and starches or spices) or meat products with an insignificant amount of fat. Another aspect is that, with the exclusion of secondary fixation, there is no necessity for the use of highly toxic OsO_4_. 

## Figures and Tables

**Figure 1 foods-09-00487-f001:**
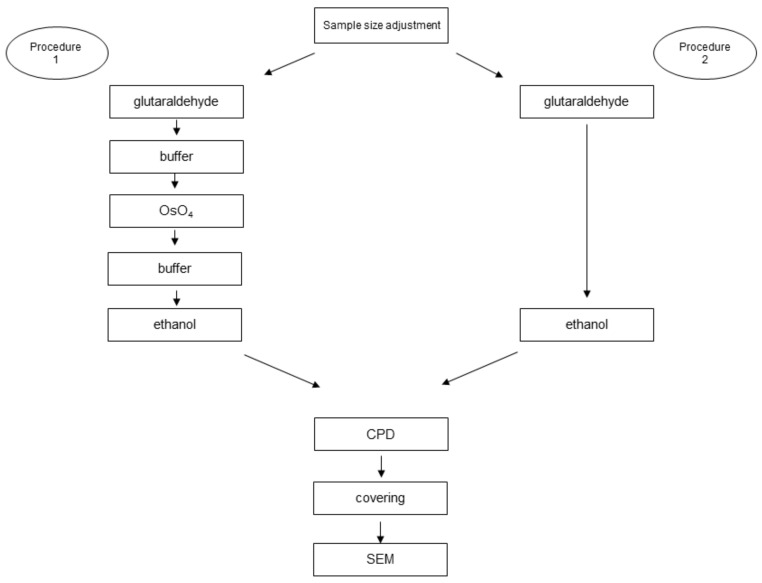
Graphic visualization of the provided methods.

**Figure 2 foods-09-00487-f002:**
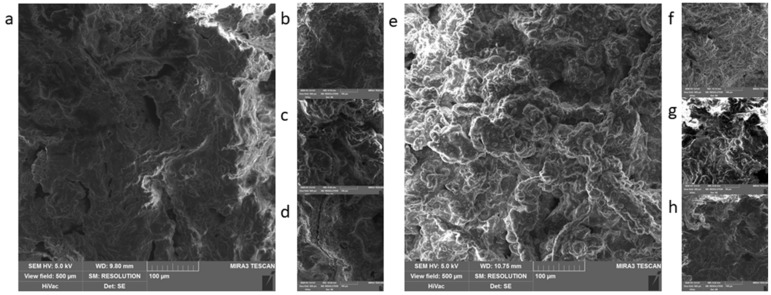
Non-heat treated meat products: (**a**,**b**)—čajovka, procedure 1; (**c**,**d**)—métský salami, procedure 1; (**e**,**f**)—čajovka, procedure 2; (**g**,**h**)—métský salami, procedure 2.

**Figure 3 foods-09-00487-f003:**
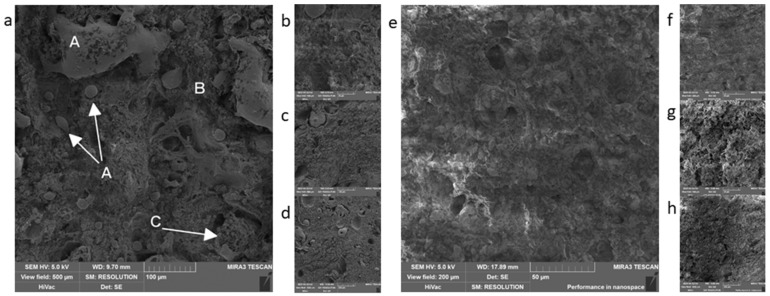
Cans: (**a**,**b**)—Játrovka, procedure 1, A—fat, B—connection (muscle), C—protein net; (**c**,**d**)—Matěj, procedure 1; (**e**,**f**)—Játrovka, procedure 2; (**g**,**h**)—Matěj, procedure 2.

**Figure 4 foods-09-00487-f004:**
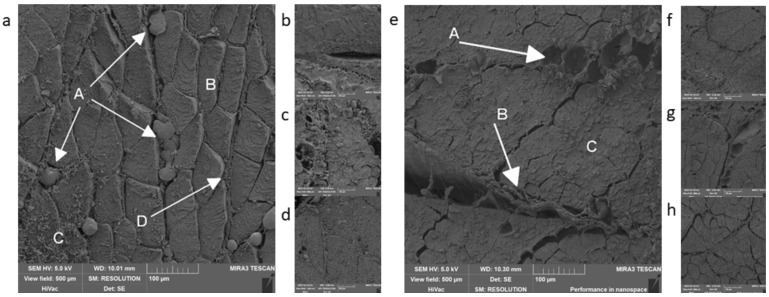
Heat-treated meat products: (**a**,**b**)—Orava bacon, procedure 1, **A**—fat droplets, **B**—muscle fibers, **C**—artifacts, **D**—connective tissue- endomysium; (**c**,**d**)—English bacon, procedure 1; (**e**,**f**)—English bacon, procedure 2, **A**—place, where there was a lipid droplet, **B**—connective tissue- perimysium, **C**—muscle fibers; (**g**,**h**)—Orava bacon, procedure 2.

**Figure 5 foods-09-00487-f005:**
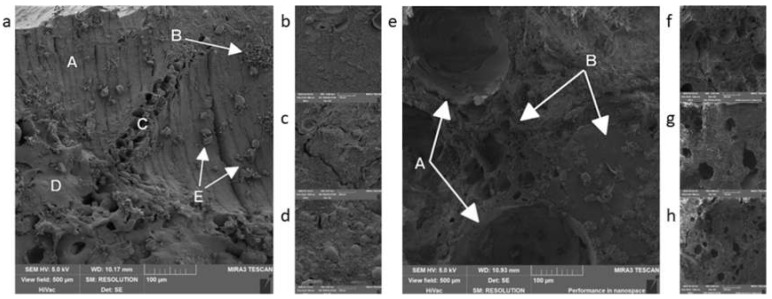
Long-life heat-treated meat products: (**a**,**b**)—“Tourist” salami, procedure 1, A—muscle fibers, B—artifacts, C—connective tissue, D—spilled, fixed fat, E—crystals- artifacts; (**c**,**d**)—“Vysočina”, procedure 1; (**e**,**f**)—“Vysočina”, procedure 2, A—empty gaps after fat droplets, B—connection; (**g**,**h**)—Tourist salami, procedure 2.

**Figure 6 foods-09-00487-f006:**
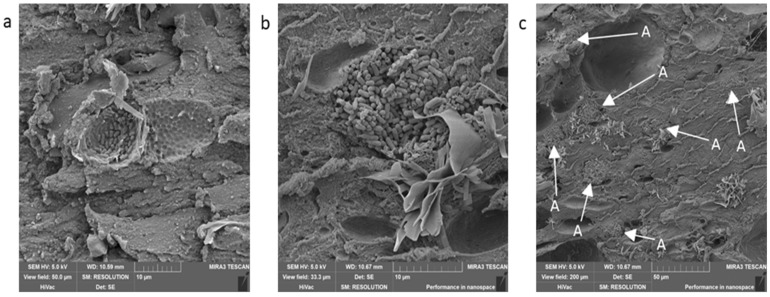
Lactic acid bacteria; (**a**)—procedure 1; (**b**,**c**)—procedure 2; A—arrangement of colonies of lactic acid bacteria.

**Figure 7 foods-09-00487-f007:**
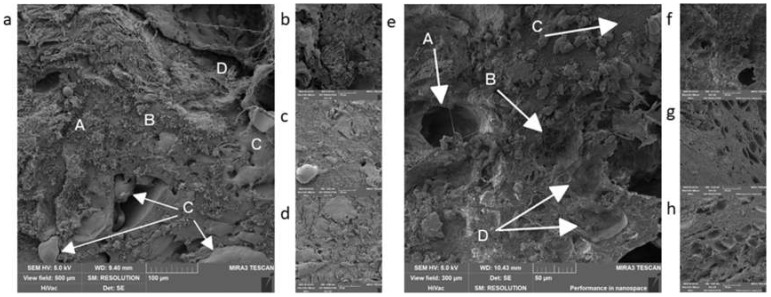
Long-life fermented meat products: (**a**,**b**)—Dunajská” sausage, procedure 1, A—artifacts, B—connection, C—fat, D—crosslinked protein; (**c**,**d**)—“Poličan”, procedure. 1; (**e**,**f**)—“Poličan”, procedure 2, A—air lacuna, B—protein net, C—connection, D—empty space after fat; (**g**,**h**)— “Dunajská” sausage, procedure 2.

**Figure 8 foods-09-00487-f008:**
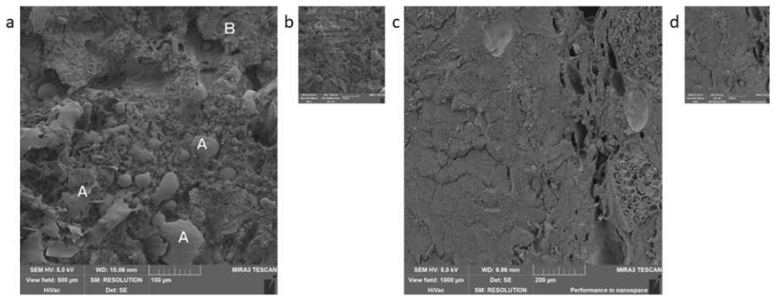
Semi-preserves: Moravian sausage mix, (**a**,**b**)—procedure 1, A—fat, B—muscle; (**c**,**d**)—procedure 2.

**Figure 9 foods-09-00487-f009:**
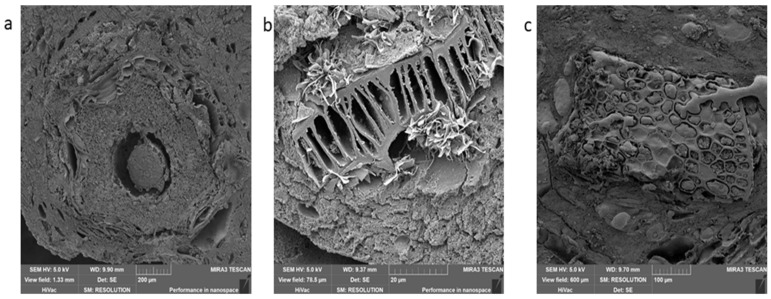
Spices found in samples: (**a**)—caraway, (**b**)—mustard, (**c**)—paprika.

**Table 1 foods-09-00487-t001:** Working procedures for product processing.

Chemicals	Concentration	Time	Procedure 1	Procedure 2
glutaraldehyde	3%	24 h	x	x
buffer-cacodylate		3 × 15 min	x	
osmium tetroxide	1%	48 h	x	
buffer-cacodylate		3 × 15 min	x	
ethanol of increasing concentrations	25%, 50%, 70%, 85%, 90%, 96%, 100%	every 15 min	x	x

x—labels the chemical, which was used in the respective procedure; the temperature during the experiment: +21 °C ± 3. Procedure 1: glutaraldehyde–cacodylate buffer–osmium tetroxide–cacodylate buffer–ethanol. Procedure 2: glutaraldehyde–ethanol.

**Table 2 foods-09-00487-t002:** Evaluation of the findings.

Type of Meat Product	Fixation Method	Containing Muscle with Visible Muscle Fibers	Space between Muscle Fibers	Shape of Muscle Fibers on Crosscut	Connective Tissue	Adipose Cells	Homogeneity	Artifacts	Other Findings
Non-heat treated	1 *	x	x	x	x	x	x	x	x
		x	x	x	x	x	x	x	x
	2 **	x	x	x	x	x	x	x	x
		x	x	x	x	x	x	x	x
Cans	1	✓	x	x	x	✓	+++	x	modified starch, air gaps
		x	x	x	x	x	+++	x	air gaps
	2	✓	+	oval	x	✓	++		air gaps, protein net
		x	x	x	x	x	++	✓	protein net, modified starch
Heat-treated	1	✓	+	round	✓	✓	++	✓	x
		✓	++	round	✓	x	++	x	x
	2	✓	+	oval	✓	✓	++	✓	x
		✓	++	oval	✓	x	++		x
Long-life heat-treated	1	x	x	x	x	✓	+++	✓✓	air gaps
		✓	x	x	x	x	+++	x	air gaps
	2	✓		oval	✓	✓	++	✓	air gaps, crystals
		x	x	x	x	x	x	x	x
Long-life fermented	1	✓	+	oval	x	✓	++	✓✓	cross-linking of proteins, spices-garlic, paprika
		✓	+	oval		x	++	✓	LAB, protein matrix-connection, spices-paprika
	2	✓	+	oval	✓	x	+	x	air gaps/lacunes
		✓	x	x	✓	✓	+	✓	LAB
Semi-preserves	1	✓	+	oval	✓	✓	++	✓	x
		✓	+	round	✓	x	+	x	spices-caraway, paprika

* Procedure 1: glutaraldehyde–cacodylate buffer–osmium tetroxide–cacodylate buffer–ethanol. ** Procedure 2: glutaraldehyde–ethanol. ✓: the visible findings; x: not visible findings; +/++/+++: quantitative findings. LAB—lactic acid bacteria.

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
