# Peer review of "Assessment of the Effect of Secondary Fixation on the Structure of Meat Products Prepared for Scanning Electron Microscopy"

_foods, 2020, doi:10.3390/foods9040487_

Round 1
Reviewer 1 Report
General comments:
This manuscript deals with a sound topic dealing with the procedures and chemicals used in the fixing of biological specimens for SEM, which is very relevant in foods domain. Moreover, the type of comparative study over a reasonably number of different meats, offered by the authors, does not appear to have been explored this way.
The manuscript is well presented and the Introduction/Results-Discussion/Conclusions sections are well-documented and explained. Authors included, as well, concise information about the different fixatives, its advantages and drawbacks.
Results are well dissected and suggestions are provided by authors in the cases were the components of meat were not recognized.
Findings (observed for meat samples in the C.R.) are very significant specially regarding the toxicity related to OsO4, and suggest a wider transposition to other meat products from different sources.
Detailed comments:
- Table 1 is elucidative about the time of soaking with each chemical. But as temperature is a relevant parameter in these methodologies, and storage conditions influence the occurrence of artfacts resulting from the exposition of samples to chemicals, what was the storage temperature in EtOH 100% until the CPD step? Moreover, no temperature information is given in the experimental section regarding the several steps of each procedure.
- In Table 1, procedures are numbered as Procedures no.1 and no.2. However, no reference to the meaning of these two procedures can be found. The only places where both procedures are explained are the Abstract and Figure 1. Why not use the same footnote appearing in Table 3?
- Table 3 should be numbered Table 2.
- Key-words should include “meat” or “meat structure” or “meat quality”.
- A good bibliographic support is provided. However, authors should carefully revise the bibliographic list in terms of the name of journals, volume, pages. Besides, authors should unify the citation of the periodicals; for example, some are fully written, some are written in abbreviation form (Ref, 20, page 14, Lines 320/321, Curr. Protocol. Microbiol. Instead of Current Protocols in Microbiology). This citation is also wrong: should be 2012, 25, 2B.2.1-2B.2.47. The citation of Ref 18 (page 14, Line 317) should be corrected: ……Microscopy Today, 8 (1), 8-13.
Reviewer 2 Report
Comments to the Author:
The manuscript reports research into assessment of the effect of secondary fixation on the structure of meat products prepared for scanning electron microscopy. The subject matter is interesting and falls into the scope of the journal. However, I suggest you to review the language and to provide the following improvements.
Line 19-20: “glutaraldehyde (GA) primary fixation” instead of “glutaraldehyde primary fixation (GA)”.
Line 20: I understand that you have used the abbreviation “GA” for the word “glutaraldehyde”. However, this abbreviation “GA” you have not used it across the manuscript, although you have used the word “glutaraldehyde”. Why don’t you use this abbreviation? If you are not going to use it across the manuscript, it is better that you remove from the abstract.
Lines 40-42: these lines correspond to the classification of meat products. But, this classification needs a reference, because there are other different classifications according to the countries and their legislation.
Line 60: I understand that you have used the abbreviation “GA” for the word “glutaraldehyde” in the abstract. Why don’t you use it in this line?
Line 75: explain why you only study one semi-preserves product.
Line 83: which is the appropriate size? Specify, please.
Line 87: specify the acronym “CPD”.
Line 99: “Table 3” instead of “Table no.3”. This comment is applicable across of the manuscript.
Table 3: there is no Table 2 in your manuscript. Thus, the Table 3 from the page 12 should be Table 2.
Line 99: you mention Table 3 in line 99, thus, the Table 3 should be in the page 5 (not in the page 12 at the final of the results and discussion section).
Line 102: “Figure 2” instead of “Figure no. 2”. This comment is applicable across of the manuscript.
Table 3:
- Explain the meaning of the symbols used in the table (✓, +, ++, x).
- Specify the acronym “LAB”.
- “Table 1” should be in bold, according to the guidelines of FOODS.
- There is no Table 2 in your manuscript. Thus, the Table 3 from the page 12 should be Table 2.
Figure 2:
- The size of the figures should be the same size. I do not understand why the size of the figures 1 and 5 bigger than the size of rest of figures.
- “Figure 2” instead of “Figure no.2”.
- “Figure 2” should be in bold, according to the guidelines of FOODS.
- The comments for figure 2 are applicable for figures 2, 3, 4, 5, 6, 7, 8 and 9.
Line 141: “Figure 4” instead of “Fig. 3”.
Line 143: “Figure” instead of “Fig.”. According to the guidelines of FOODS, “all figures and tables should be cited in the main text as Figure 1, Table 1, etc.”. Thus, this comment is applicable across the manuscript. From now on, I am not going to indicate this issue any more, but you should correct this issue across the manuscript because you have several mistakes related to this issue.
Line 144: “procedure 1” instead of “procedure no. 1”. From now on, I am not going to indicate this issue any more, but you should correct this issue across the manuscript because you have several mistakes related to this issue.
Line 206: “method 1” instead of “method no. 1”. From now on, I am not going to indicate this issue any more, but you should correct this issue across the manuscript because you have several mistakes related to this issue.
Lines 228-230: correct the title of the Figure 8.
The “Result and Discussion” section needs to be re-organized for clarity.
Lines 271-272: you should remove these lines because they are from “foods-template” file.
Lines 279-281: you should remove these lines because they are from “foods-template” file.
Lines 282: you should remove this line because they are from “foods-template” file.
References: there are mistakes in almost all references. You should correct the references according to the guidelines of FOODS.
Author Response
Prosím podívejte se na přílohu

Round 2
Reviewer 2 Report
Comments to the Author:
I suggest you to provide the following improvements.
Figure 2: The size of the figures should be the same size. I do not understand why the size of the pictures 1 and 5 (from figure 2, which contains 8 pictures) is bigger than the size of rest of pictures.
This comment for figure 2 is applicable for figures 2, 3, 4, 5, 7 and 8.
Author Response
Dear Reviewer, We finally managed to make pictures in figures to have the same size. We hope that now it will be satisfactory. Best regards, Hana Běhalová
